evolution/genomics/behaviour

gene regulation, small non-coding RNA, microRNA targets, eusociality, lineage-specific

**Author for correspondence:**
Karen M. Kapheim
e-mail: karen.kapheim@usu.edu

# Brain microRNAs among social and solitary bees

Karen M. Kapheim[1], Beryl M. Jones[2], Eirik Søvik[3], Eckart Stolle[4], Robert M. Waterhouse[5], Guy Bloch[6] and Yehuda Ben-Shahar[7]

[1]Department of Biology, Utah State University, 5305 Old Main Hill, Logan, UT 84322, USA
[2]Department of Ecology and Evolutionary Biology, Princeton University, Princeton, NJ 08544, USA
[3]Department of Science and Mathematics, Volda University College, 6100 Volda, Norway
[4]Centre of Molecular Biodiversity Research, Forschungsmuseum Alexander Koenig, Adenauerallee 160, 53113 Bonn, Germany
[5]Department of Ecology and Evolution, University of Lausanne and Swiss Institute of Bioinformatics, 1015 Lausanne, Switzerland
[6]Department of Ecology, Evolution and Behavior, The Alexander Silberman Institute of Life Sciences, The Hebrew University of Jerusalem, Jerusalem 91904, Israel
[7]Department of Biology, Washington University in St Louis, St Louis, MO 63130, USA

 KMK, 0000-0002-8140-7712; RMW, 0000-0003-4199-9052;
GB, 0000-0003-1624-4926

Evolutionary transitions to a social lifestyle in insects are associated with lineage-specific changes in gene expression, but the key nodes that drive these regulatory changes are unknown. We examined the relationship between social organization and lineage-specific microRNAs (miRNAs). Genome scans across 12 bee species showed that miRNA copy-number is mostly conserved and not associated with sociality. However, deep sequencing of small RNAs in six bee species revealed a substantial proportion (20–35%) of detected miRNAs had lineage-specific expression in the brain, 24–72% of which did not have homologues in other species. Lineage-specific miRNAs disproportionately target lineage-specific genes, and have lower expression levels than shared miRNAs. The predicted targets of lineage-specific miRNAs are not enriched for genes with caste-biased expression or genes under positive selection in social species. Together, these results suggest that novel miRNAs may coevolve with novel genes, and thus contribute to lineage-specific patterns of evolution in bees, but do not appear to have significant influence on social evolution. Our analyses also support the hypothesis that many new miRNAs are purged by selection due to deleterious effects on mRNA targets, and suggest genome structure is not as influential in regulating bee miRNA evolution as has been shown for mammalian miRNAs.

# 1. Introduction

Eusociality has evolved several times in the hymenopteran insects. In its most basic form, this lifestyle involves reproductive queens living with their worker daughters who forgo direct reproduction to cooperatively defend the nest, care for their siblings and forage for the colony. Due to the complex nature of this lifestyle, the evolution of eusociality probably requires modification of molecular pathways related to development, behaviour, neurobiology, physiology and morphology [1]. The evolution of eusociality is thus expected to involve both genetic changes as well as changes in the way the genome responds to the environment [2]. Recent studies have found that social insect species share evolutionary genomic changes that may reflect an increased capacity for gene regulation [3,4]. Evidence for this comes from signatures of rapid evolution of genes involved in transcription and translation, gene family expansions of transcription factors, and increasing potential for transcription factor binding activity in conserved genes. Interestingly, while these types of regulatory changes are common to independent origins and elaborations of eusociality, the specific genes and regulatory elements involved are unique to each lineage [3–5]. This suggests that lineage-specific processes are influential in generating new patterns of gene regulation that contribute to social behaviour.

Small, non-coding RNAs such as microRNAs (miRNAs) may be an important source of regulatory novelty associated with the evolution of phenotypic complexity, including eusociality. MiRNAs are short (approx. 21–22 nt), non-coding RNAs that regulate protein-coding genes through post-transcriptional binding to the 3′ UTR of messenger RNA (mRNA) transcripts, in most cases preventing translation or causing mRNA degradation [6]. Each miRNA can target dozens to hundreds of mRNAs, and may therefore regulate multiple gene networks [6,7]. Like mRNAs, miRNAs are spatially and temporally specific in their expression patterns. Thus, complex changes in gene regulation can be achieved with relatively minor changes in miRNA expression. This can result in major phenotypic shifts or fine-tuning of phenotypic optimization [6]. Novel miRNAs originate in a variety of genomic features, including exons and introns of protein-coding and non-coding RNA genes, transposable elements, pseudogenes or intergenic regions, and thus emerge and disappear over relatively rapid timescales [8–11]. It is thus not surprising that expansion of the miRNA repertoire is associated with the evolution of morphological complexity across the tree of life [9,12,13].

There is accumulating evidence for a role of miRNAs in regulating the social lives of insects. While most miRNAs seem to be conserved in major lineages of insects [14,15], expression levels vary across individuals performing different social functions, such as between workers performing different tasks in honeybees [16–18]. MiRNAs may also play a role in caste determination, as queen- and worker-destined larvae express different sets of miRNAs throughout development in honeybees [19–21] and bumblebees [22]. Additionally, miRNAs play a role in regulating some physiological correlates of social behaviour in honeybees, including activation of ovaries in queens and workers [23] and response to the reproductive protein *vitellogenin* [24]. Together, these studies suggest that miRNAs could play a role in the evolution of eusociality through their effects on gene regulatory networks involved in socially relevant traits. A rigorous test of this hypothesis requires comparisons of the presence, expression and function of miRNAs across related species that vary in social organization.

Here, we present a comparative analysis of miRNAs across bee species with variable social organization. We first looked for miRNA repertoire expansions associated with eusociality by scanning 12 bee genomes for known miRNAs, and statistically evaluating copy-number of each miRNA type with regard to differences in sociality in a phylogenetic model. We then described and compared miRNAs expressed in the brains of six bee species from three families that include repeated origins of eusociality. We tested the hypothesis that changes in gene regulatory function associated with social evolution are facilitated by lineage-specific miRNAs with two predictions: (i) If lineage-specific miRNAs are assimilated into ancestral gene networks, their predicted target genes should be ancient and conserved. (ii) If lineage-specific miRNAs play a role in social evolution, their predicted targets should be enriched for genes associated with social behaviour (e.g. caste-biased expression) or genes that are under selection in social species. We do not find evidence for a role of lineage-specific miRNAs in social evolution. However, we do identify unexpected patterns of coevolution between miRNAs and their putative target genes. We interpreted our results in light of current hypotheses for patterns of miRNA evolution in vertebrates.

# 2. Material and methods

## 2.1. microRNA diversification

We performed genome scans for small RNAs across 12 bee genomes (electronic supplementary material, table S1) using covariance models implemented with Infernal (v. 1.1) cmsearch using the gathering

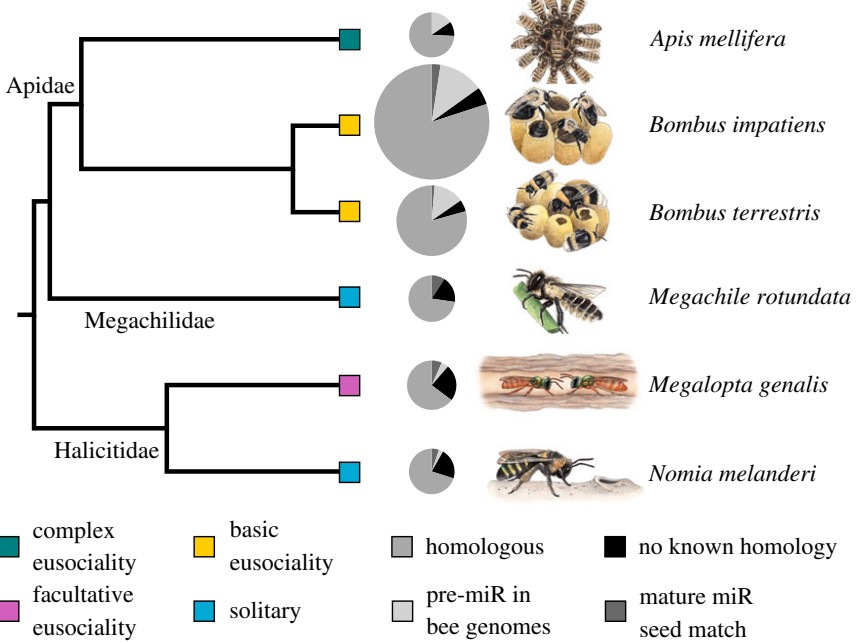

**Figure 1.** Diversity of miRNAs expressed in the brains of six bee species. The three types of homology (shades of grey) correspond to those in table 1. Black has not been previously detected in other species. Pie size corresponds to number of miRNAs detected from small RNA sequencing. Boxes indicate social organization (green, complex eusociality; yellow, basic eusociality; pink, facultative eusociality; blue, solitary). Phylogenetic relationships are following previous studies [28–30].

threshold for inclusion (–cut_ga) [25] to find all Rfam (release 29.0) accessions in each genome. We used Spearman rank regressions to test for associations between miRNA copy-number and social biology. We categorized each species as solitary, facultative basic eusocial, obligate basic eusocial or obligate complex eusocial following Kapheim *et al.* [4]. We used the ape package (v. 3.1) [26] in R (v. 3.5.0) [27] to calculate phylogenetic independent contrasts for both social organization and miRNA copy-number, cor.test to implement Spearman's rank correlations and p.adjust with the Benjamini–Hochberg method to correct for multiple comparisons.

## 2.2. Sample acquisition, RNA isolation and sequencing

We used the brain from a single adult female from six bee species, including both eusocial and solitary species with well-studied behaviour from three bee families (figure 1). Details of sample collection, RNA isolation and sequencing are provided in electronic supplementary material, table S2.

## 2.3. microRNA discovery and quantification

We used miRDeep2 (v. 2.0.0.8) [31] to identify and quantify miRNAs expressed in the brains of each species, with a three-step process of miRNA detection to identify homologous miRNAs between species. First, we gathered mature miRNA sequences previously described in other insect species (electronic supplementary material, table S3). Reads for each sample were quality filtered (minimum length 18, removal of reads with non-standard bases), adapter-trimmed, and aligned to the species' genome (electronic supplementary material, table S1) with the mapper.pl script. Approximately 61–84% of reads successfully mapped (electronic supplementary material, table S2).

We then identified known and novel miRNAs in each sample with the miRDeep2.pl script, using our set of insect miRNAs (electronic supplementary material, table S3) as known mature sequences. The quantifier.pl script generated sets of known and novel miRNAs in each sample, along with quantified expression information. We filtered novel miRNAs in each species according to the following criteria: no rRNA/tRNA similarities, minimum of five reads each on mature and star strands of the hairpin sequence, and a randfold $p$-value < 0.05. Randfold describes the RNA secondary structure of potential precursor miR (pre-miRs) [31].

We used these filtered miRNAs in a second run of detection and quantification, repeating the pipeline above after adding mature sequences of novel miRNAs from each species to our set of known miRNAs. This allowed detection of homologous miRNAs (based on matching seed sequences) not represented in miRBase across our species. We applied the same set of filtering criteria as above.

Some novel miRNAs may exist in the genomes of other bees, even if they are not expressed. We used blastn (-perc_identity 50 -evalue 1e-5) to search for homologous pre-miR sequences in 12 bee genomes (electronic supplementary material, table S1) for each novel miRNA without a matching seed sequence.

## 2.4. microRNA localization

We used bedtools (v. 2.27.0) intersect [32] to find overlap of miRNAs with predicted gene models (electronic supplementary material, table S4), and repetitive element annotations from previously established repeat libraries that had been generated using Repeatmasker [4,28,33–36].

## 2.5. Target prediction

We extracted potential target sites 500 bp downstream from each gene model using bedtools flank and getfasta [32], following previous studies [19] and an average 3′ UTR of 442 nt in *Drosophila melanogaster* [37]. Target prediction was run with miRanda (v. 3.3) [38] (minimum energy threshold −20, minimum score 140, strict alignment to seed region [-en -20 -sc 140 –strict]) and RNAhybrid (v. 2.12) [39] (minimum free energy threshold −20). We kept only miRNA-target gene pairs that were predicted by both programs with $p < 0.01$.

## 2.6. Target age and functional enrichment

Gene ages were determined using orthogroups from OrthoDB (v. 9) [40], which includes *Apis mellifera*, *Bombus impatiens*, *Bombus terrestris*, and *Megachile rotundata*. Gene sets of *Megalopta genalis* and *Nomia melanderi* were mapped to Metazoa-level (330 species) orthogroups. Gene ages were inferred from the taxonomic breadth of all species in each orthogroup, with at least one representative from each of the following groups which does not belong to the next lower group: Vertebrata, Metazoa, Arthropoda, Insecta, Holometabola, Hymenoptera, Aculeata, Apoidea. Genes without identifiable orthologues were labelled 'Unique'.

## 2.7. Enrichment tests of lineage-specific miRNA targets

For each species, gene expression datasets related to socially relevant phenotypes (e.g. caste) were compared against targets of lineage-specific miRNAs (electronic supplementary material, table S5). For *M. genalis* caste data, RNAseq reads from Jones *et al.* [41] (NCBI PRJNA331103) were trimmed using Trimmomatic (v. 0.36) [42] and aligned to a draft genome assembly of *M. genalis* (NCBI PRJNA494872) [35] using STAR (v. 2.5.3) [43]. Gene counts were obtained using featureCounts in the Subread package (v. 1.5.2) [44], and differential expression analysis was conducted using edgeR [45] as in Jones *et al.* [41].

We also tested datasets identifying genes under selection in bee species [34,46,47] or across social lineages of bees [4,48] for enrichment of lineage-specific miRNA targets (electronic supplementary material, table S5). When necessary, we used reciprocal blastp (evalue $< 10e^{-5}$) to identify orthologous genes, and only genes with putative orthologues were included. Hypergeometric tests (using phyper in R) were used to test for over- or under-enrichment between each pair of lists. The representation factor (RF) given represents the degree of overlap relative to random expectation (RF = 1). RF is calculated as RF = $x/E$, where $x$ is the number of genes in common between two lists and $E$ is the expected number of shared genes. ($E = nD/N$, where $n$ is the number of genes in list 1, $D$ is the number of genes in list 2, and $N$ is the total number of genes.)

# 3. Results

## 3.1. Low levels of microRNA copy-number variation among bee genomes

Our genome scans revealed very little variation in copy-number of most miRNAs. Of the 50 miRNA Rfam accessions, half had the same number of copies (1 or 2) in all 12 bee genomes (electronic supplementary

material, table S6). The mean copy-number across all miRNAs in all bee genomes was $1.19 \pm 0.74$. One exception was miR-1122, for which we found 70 copies in *M. genalis*, but no copies in the other species. We did not find any significant associations between miRNA copy-number and social organization (electronic supplementary material, table S6).

## 3.2. Expressed microRNA diversity in bee brains

We identified 97–245 known and novel miRNAs expressed in the brains of each of our six species (electronic supplementary material, table S7). The majority of these were intergenic or within introns (table 1). Each species had at least one miRNA originating from exons of protein-coding genes and repetitive DNA (table 1). Most of the overlap between miRNA precursors and repetitive DNA corresponded to uncharacterized repeat elements, with few overlaps with well-characterized transposons or retrotransposons (table 1). Variation in number of expressed miRNAs in each species was not related to observable technical variation, such as sequencing centre, number of reads, number or proportion of reads mapped to the genome or type of sample from which they were obtained (electronic supplementary material, table S2). This variation in number of expressed miRNAs is similar to that found in other groups of species with shorter divergence times [49].

Most detected miRNAs in each species had known homologues in at least one other species. However, each species had a substantial proportion (20–35%) of detected miRNAs with lineage-specific expression in the brain (table 1 and figure 1), 24–72% of which did not have any known homologues in other species (table 1). We defined lineage-specific miRNAs as those with lineage-specific expression and with no seed match to a known mature miRNA (table 1, columns 6–7), because these show the most evidence of being real miRNAs that are unique to a particular species. (Sequence similarity of pre-miRs in other bee genomes is not evidence that a mature miRNA is transcribed.) Lineage-specific miRNAs had significantly lower expression levels compared with homologous miRNAs in each species (*t*-tests: *A. mellfera*, *M. rotundata*, *M. genalis* $p < 0.001$, *B. impatiens*, *B. terrestris* $p < 0.01$, *N. melanderi* $p < 0.05$).

Lineage-specific miRNAs were localized both within genes and intergenically. The proportion of lineage-specific miRNAs that were intra- or intergenic was similar to miRNAs with homologues for every species except *N. melanderi*, for which a disproportionate number of lineage-specific miRNAs were intragenic ($\chi^2 = 4.78$, $p = 0.03$). Genes that serve as hosts for intragenic lineage-specific miRNAs were not significantly older than would be expected by chance (i.e. belong to orthogroups shared with vertebrates) in any species (hypergeometric tests: $p = 0.14$–$0.76$). Across all species, genes serving as hosts for intragenic lineage-specific miRNAs were not significantly older than genes hosting miRNAs with known homologues ($\chi^2$ tests: $p = 0.05$–$0.89$).

Of miRNAs with homologues, most were expressed in all six species, but one miRNA (miR-305) was expressed in the brains of each of the social, but not the solitary, species (figure 2). Although we did not detect expression of miR-305 in the two solitary species, *M. rotundata* and *N. melanderi*, genome scans of each species against the Rfam database suggested all bee species have one copy of miR-305 (electronic supplementary material, table S6). Predicted targets of miR-305 differed across species. *Oxysterol* (OG EOG091G0FV2) was a common target among the (social) Apidae bees, but was not among the targets for *M. genalis*. However, *arylformamidase* (OG EOG091G0KT8), which is also involved in lipid metabolism and transport, was a predicted target in *M. genalis*. *Synaptobrevin* (OG EOG091G0MPE), which is involved in synaptic plasticity and neurotransmitter release, was a predicted target of miR-305 in *B. impatiens*.

## 3.3. Lineage-specific microRNAs preferentially target lineage-specific genes, but not genes with caste-biased expression or genes under positive selection

If lineage-specific changes in gene regulatory function associated with social evolution are facilitated by novel miRNAs inserted into existing gene networks, then predicted targets of lineage-specific miRNAs should be highly conserved and enriched for genes with known functions in social evolution. Most predicted mRNA targets of lineage-specific miRNAs were highly conserved and belonged to orthogroups shared by vertebrates (figure 3; electronic supplementary material, table S8), but not significantly more than expected given the large number of conserved genes in each genome (hypergeometric tests: $p > 0.99$). We did, however, find significant enrichment for genes unique to each species among the predicted targets of lineage-specific miRNAs (hypergeometric tests: *A. mellifera*: $RF = 1.51$, $p = 5.44e^{-5}$; *B. impatiens*: $RF = 1.28$, $p = 0.02$; *B. terrestris*: $RF = 1.78$, $p = 1.90e^{-6}$; *M. rotundata*: $RF = 1.79$, $p = 0.0002$; *M. genalis*: $RF = 1.62$, $p = 1.48e^{-12}$; *N. melanderi*: $RF = 1.78$, $p = 9.02e^{-5}$), indicating

**Table 1.** Localization of miRNAs in the genomes of six bee species. Numbers not in parentheses represent features on the same strand as the pre-miR. Numbers in parentheses indicate strand mismatch. Some pre-miRs overlapped with one or more genes on both the same and opposite strands, and are thus counted twice (*A. mellifera* and *M. genalis*, 1; *B. impatiens*, 5; *B. terrestris*, 4; *N. melanderi*, 3). Seed match—mature miR had a seed match with a known miR; pre-miR—successful blastn hit to the pre-miR sequence in at least one other bee genome; unique—no homologue was found in other species (seed match to mature or blastn hit to pre-miR).

| species | sociality | expressed miRs | miRs with lineage-specific expression in the brain | | | | location in the genome | | | | |
| --- | --- | --- | --- | --- | --- | --- | --- | --- | --- | --- | --- |
| | | | total | seed match | pre-miR | unique | intergenic | exon | intron | transposable element | uncharacterized repetitive DNA |
| *Apis mellifera* | complex eusocial | 97 | 25 | 0 | 15 | 10 | 45 | 5 | 38 (10) | 0 | 0 |
| *Bombus impatiens* | basic eusocial | 245 | 49 | 6 | 31 | 12 | 129 | 4 (1) | 89 (27) | 7 | 32 |
| *Bombus terrestris* | | 150 | 31 | 2 | 21 | 8 | 76 | 1 (1) | 56 (20) | 13 | 36 |
| *Megalopta genalis* | facultative eusocial | 105 | 37 | 7 | 5 | 25 | 63 | 3 | 30 (10) | 2 | 28 |
| *Megachile rotundata* | solitary | 99 | 27 | 9 | 0 | 18 | 48 | 8 (1) | 37 (5) | 2 | 15 |
| *Nomia melanderi* | solitary | 97 | 29 | 5 | 3 | 21 | 50 | 8 | 34 (8) | 2 | 27 |

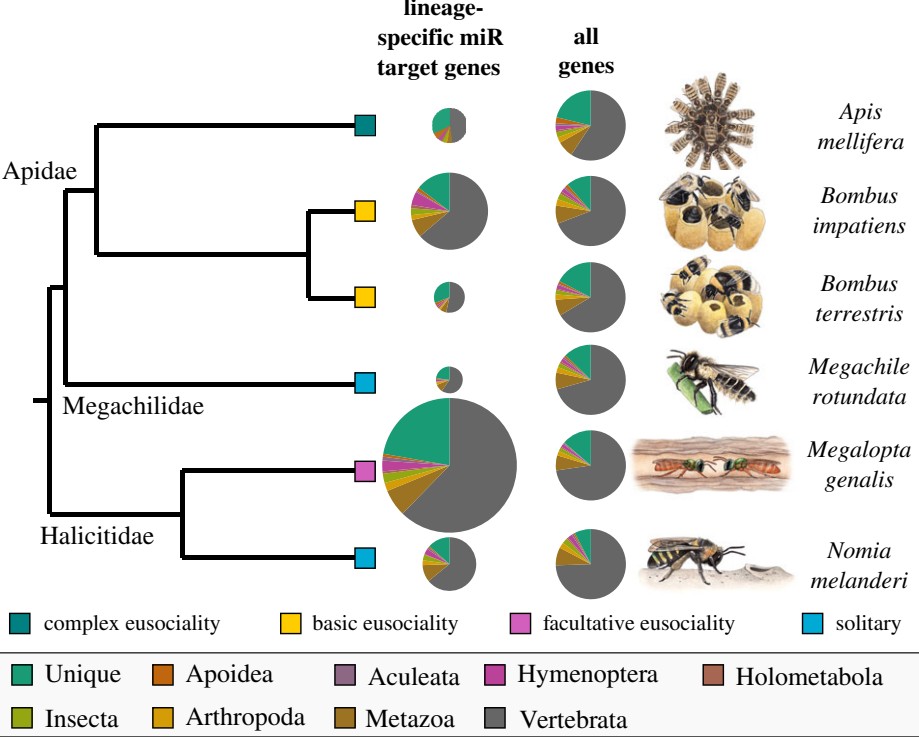

**Figure 2.** Overview of miRNAs shared by one or more of six bee species. Dots connected by lines indicate which species each set of miRNAs is shared between. The size of each miRNA set is indicated above the corresponding bar. The grey bar highlights that only 30 miRNAs were expressed in the brains of all six species. The arrow labels miR-305, the only miRNA expressed in the brains of all social species, but none of the solitary species.

**Figure 3.** Age of genes targeted by lineage-specific miRNAs. Genes predicted to be targeted by lineage-specific miRNAs are more likely to be unique to each species than predicted by chance. Pie chart size is scaled to number of predicted target genes for lineage-specific miRNAs, but not for all genes. Colour slices indicate orthogroup age for each predicted gene. The green slice (lineage-specific genes) is larger for the set of genes predicted to be targeted by lineage-specific miRNAs than for all genes.

that novel miRNAs are more likely to target novel genes than would be expected by chance (figure 3; electronic supplementary material, table S8).

We did not find support for the prediction that novel miRNAs should target genes that function in social behaviour and evolution. We first considered the genes that are differentially expressed between castes in social species, because these are likely to be involved in regulating behavioural and physiological aspects of sociality. Predicted targets of lineage-specific miRNAs were not significantly enriched for genes with caste-biased expression in the social species (electronic supplementary material, figure S1 and table S5). Also contrary to our prediction, targets of lineage-specific miRNAs were not enriched for genes under positive selection in any species (electronic supplementary material, figure S1 and table S5). In fact, genes under positive selection in the halictid bees were significantly depleted for targets of lineage-specific miRNAs (hypergeometric tests: *M. genalis*: RF = 0.3, $p = 3.72e^{-7}$; *N. melanderi*: RF = 0.3, $p = 9.79e^{-4}$). We also assessed overlaps with genes previously found to be under positive selection in social species, compared with solitary species [4,48], but found no significant overlap or depletion with predicted targets of lineage-specific genes (hypergeometric tests: $p > 0.05$; electronic supplementary material, figure S1 and table S5).

## 4. Discussion

Eusociality is a major evolutionary innovation that requires regulatory changes in a wide range of molecular pathways [1]. We tested the hypothesis that miRNAs play a role in the evolution of eusociality via their regulatory effects on gene networks by comparing miRNA expression in a single bee from three eusocial and three solitary species that span three families. Our results provide very limited support for this hypothesis.

We identified a single miRNA (miR-305) that was expressed exclusively in the brains of social bees in our study. The presence of this miRNA in the solitary bee genomes suggests that an evolutionary shift in expression pattern may have accompanied at least two independent origins of eusociality in bees. This miRNA coordinates Insulin and Notch signalling in *D. melanogaster*, both of which are important regulators of social dynamics in insects [50–52]. Interestingly, miR-305 is also upregulated in worker-destined compared with queen-destined honeybee larvae, and may thus play a role in caste differentiation [20]. Further investigation with deeper sampling and additional social and solitary species is necessary to determine whether this miRNA is expressed exclusively in the brains of social species and how it may influence social behaviour.

We focused attention on miRNAs for which no mature miRNAs with seed matches were detected in any other species, because these may influence the lineage-specific patterns of gene regulatory changes previously shown to influence social evolution [3,4]. We hypothesized that if novel miRNAs are inserted into existing gene networks that become co-opted for social evolution, they should target genes that are highly conserved. Instead, we find that targets of lineage-specific miRNAs are enriched for lineage-specific genes, while genes belonging to ancient orthogroups were not more likely to be targets than expected by chance. This suggests that novel miRNAs coevolve with novel genes, as has been shown for the evolution of cognitive function in humans [53]. Previous work in honeybees has shown that taxonomically restricted genes play an important role in social evolution, with expression of these genes biased toward glands with specialized functions for life in a social colony (e.g. the hypopharyngeal and sting glands) [54], and upregulated in workers [55]. Thus, it is reasonable to expect that new miRNAs targeting new genes could have important social functions.

Alternatively, it is possible that new miRNAs targeting lineage-specific genes are transient and will be purged by natural selection because they are less integrated into existing gene networks [10,56,57]. Emergent miRNAs are expected to initially have limited expression to mitigate potential deleterious effects on their target genes. Thus, lineage-specific miRNAs with low levels of expression may be in the process of being purged and may not have accumulated gene targets with important functions [9,10]. Evidence for this model comes from primates [58] and flies [11,59]. Likewise, we find that lineage-specific miRNAs have reduced expression compared with those with homologues. A purging process could explain why there are large differences in the numbers of miRNAs detected in even closely related species (e.g. the two *Bombus* species), though this could also be an effect of limited sampling. Functional analysis of lineage-specific genes in additional tissues and life stages will help to resolve their roles in social evolution.

We do not find support for the prediction that lineage-specific miRNAs should target genes associated with caste in social bees. Consistent with this observation, regulatory relationships between miRNAs and genes with caste-biased expression were not found among two other social insect species [60]. Previous studies have identified miRNAs that are differentially expressed between queens

and workers in honeybees [19–21] and bumblebees [22]. However, without comparison with other bee species, it was unknown if these caste-biased miRNAs were unique to social species. Our results suggest this is not the case. This is perhaps unsurprising in light of our finding that lineage-specific miRNAs target an unexpectedly high proportion of lineage-specific genes, potentially through coevolution. Although lineage-specific genes play an important role in sociality [61], most caste-biased genes belong to highly conserved molecular pathways [62].

Lineage-specific miRNAs also showed no evidence for preferential targeting of genes under positive selection—either within or across species. By contrast, we find these emergent miRNAs are less likely than expected by chance to target genes under positive selection in the two halictid bees. A potential explanation is that genes adaptively targeted by miRNAs tend to be under purifying selection to maintain regulatory relationships with their targets, preventing gene mis-expression [63–65]. This selective constraint is likely to be most significant in the 3′ UTR, where miRNA binding sites are located.

A more likely explanation for both of these negative results involves the hypothesized pattern of miRNA origins and assimilation [10]. This model suggests that new miRNAs are likely to have many targets throughout the genome due to chance. Most initial miRNA-target regulatory relationships are likely to have slightly deleterious effects, and would be quickly purged through purifying selection. These deleterious effects could be particularly strong for target genes with caste-biased expression or undergoing positive selection, because changes in the functional regulation of these genes are likely to have significant fitness consequences. Also, genes with caste-biased expression and those under positive selection are undergoing rapid evolution [66], and thus may be more likely to 'escape' control by errant miRNAs. Indeed, it is easier for mRNAs to lose miRNA target binding sites, which typically require exact sequence matches, than to gain them [10]. Thus, emergent miRNAs may not be expected to target adaptively or fast evolving genes, regardless of their role in social evolution.

Our analyses reveal important differences in patterns of miRNA evolution between bees and other species. For example, expansion in miRNA repertoire is associated with the evolution of animal complexity in a wide range of species [9,12,13]. The evolution of eusociality from a solitary ancestor is associated with increases in phenotypic complexity, and considered to be one of the major transitions in evolution. We therefore hypothesized that evolutionary increases in social complexity would be associated with expansions in the number of miRNAs found within bee genomes. On the contrary, we find that most bees have a single copy of previously identified miRNAs in their genomes, consistent with results of comparative genome scans across ants [3]. A recent study of miRNA diversity in insects found that morphological innovations such as holometabolous development was accompanied by the acquisition of only three miRNA families [15]. This suggests that insect evolution is not as reliant on major expansions of miRNA families as other taxonomic groups.

Additionally, our characterization of lineage-specific miRNAs expressed in the brain of each species reveals that genome structure is not as influential in regulating bee miRNA evolution as has been shown for human miRNAs. Novel human miRNAs tend to arise within ancient genes that have multiple functions and broad expression patterns, which may facilitate persistence of emergent miRNAs by increasing their expression repertoire [56,57]. In our study, lineage-specific miRNAs did not differ from previously identified miRNAs in their genomic locations in all but one species (N. melanderi). We also do not find a consistent pattern between new miRNAs and host gene age, even though a similar proportion of bee miRNAs are located within introns (31–43%; table 1), compared with in vertebrates (36–65%) [8]. However, the fact that 73–88% of bee miRNAs localized to genes are encoded on the sense strand suggests that they would benefit from host transcription, as is observed in vertebrates [8]. Additional research with insects will be necessary to identify general patterns of miRNA evolution in relationship to genome structure.

Our study identifies patterns of miRNA evolution in a set of bees that vary in social organization, and highlights important similarities and differences in the emergence patterns and functions of mammalian and insect genomes. We find no evidence that emergent miRNAs function in lineage-specific patterns of social evolution, but we do find evidence of potential coevolution of novel miRNAs and species-specific targets. We do not see an overall increase in the number of miRNAs in the genome or expressed in the brains of species with more complex eusociality. However, we do find one miRNA (miR-305) expressed in the brains of social, but not solitary, species. Empirical tests of miRNA function across additional species with variable social organization will further improve our understanding of how gene regulatory evolution gives rise to eusociality.

Ethics. Permission to collect samples of *Megalopta genalis* in Panama was granted by the Smithsonian Tropical Research Institute, and samples were exported under permit SEX/A-37-15. Permission to collect samples of *Nomia melanderi* was granted by private landowners.

Data accessibility. Sequences are deposited at NCBI SRA as BioProject PRJNA559906. Other data supporting this article are available in electronic supplementary material, tables S1–S8. Relevant code for this research is stored in GitHub: www.github.com/kapheimlab/bee_microRNA and have been archived within the Zenodo repository: https://zenodo.org/badge/latestdoi/250347886.

Authors' contribution. K.M.K. conceived of the study and designed the experiments. K.M.K., E.S., G.B. and Y.B.-S. collected the data. K.M.K., B.M.J., E.S. and R.M.W. analysed the data. K.M.K. wrote the initial draft of the manuscript. All authors edited and approved the final article for publication. All authors agree to be held accountable for the work performed therein.

Competing interests. The authors declare no competing interests.

Funding. Financial support came from the USDA National Institute of Food and Agriculture (2018-67014-27542 to K.M.K.); the Utah Agricultural Experiment Station, Utah State University (Project 1297, journal paper number 9239 to K.M.K.); the U.S.-Israel Binational Science Foundation (BSF 2012807 to G.B. and Y.B.-S.) and the Swiss National Science Foundation (PP00P3_170664 to R.M.W.).

Acknowledgements. Sequencing was performed at the University of Illinois Roy J. Carver Biotechnology Center. We thank the University of Utah High Performance Computing Center for computational time and assistance. Illustrations were created by J. Johnson (LifeSciences Studios). G.B. thanks the Clark Way Harrison Visiting Professor in Arts and Sciences that supported his stay in Washington University in St Louis. We thank G. Robinson for helpful feedback on an earlier draft of this manuscript.

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
