## [Reviewer comments · Royal Society Open Science]

Review History

Decision letter (RSOS-200517.R0)

Dear Dr Kapheim

On behalf of the Editors, I am pleased to inform you that your Manuscript RSOS-200517 entitled "Brain microRNAs among social and solitary bees" has been accepted for publication in Royal Society Open Science subject to minor revision. The handling editors have recommended publication, but also suggest some minor revisions to your manuscript. Please find the Associate Editor comments at the end of this email. Please revise the manuscript to take account of his comment concerning a note on low sample size in the Discussion.

- Ethics statement

- Data accessibility

<http://datadryad.org/submit?journalID=RSOS&manu=RSOS-200517>

- Competing interests

- Authors' contributions

- Acknowledgements

- Funding statement

Because the schedule for publication is very tight, it is a condition of publication that you submit the revised version of your manuscript before 07-Jun-2020. Please note that the revision deadline will expire at 00.00am on this date. If you do not think you will be able to meet this date please let me know immediately.

To revise your manuscript, log into <https://mc.manuscriptcentral.com/rsos> and enter your Author Centre, where you will find your manuscript title listed under "Manuscripts with

Decisions". Under "Actions," click on "Create a Revision." You will be unable to make your revisions on the originally submitted version of the manuscript. Instead, revise your manuscript and upload a new version through your Author Centre.

If your manuscript is newly submitted and subsequently accepted for publication, you will be asked to pay the article processing charge, unless you request a waiver and this is approved by Royal Society Publishing. You can find out more about the charges at <https://royalsocietypublishing.org/rsos/charges>. Should you have any queries, please contact openscience@royalsociety.org.

Kind regards,

on behalf of Dr Rees Kassen (Associate Editor) and Steve Brown (Subject Editor)
openscience@royalsociety.org

Associate Editor Comments to Author (Dr Rees Kassen):

This paper presents an interesting set of - largely negative - results around the putative association between miRNA regulation and sociality in bees. In brief, there doesn't seem to be any compelling link that miRNA evolution is linked to sociality. This is an interesting result and I commend the authors for revising their manuscript to reflect this conclusion more clearly. In light of these revisions, which are based on comments from an AE and three reviewers, I think the results are sound and the conclusions warranted. The one caveat, which deserves mention in the discussion, is the very limited sample size. This may be an inevitable consequence of the challenge of getting this kind of data but it is important to acknowledge explicitly, as the results could be different with broader sampling. Otherwise I am happy to recommend publication.

Author's Response to Decision Letter for (RSOS-200517.R0)

See Appendix A.

Decision letter (RSOS-200517.R1)

Dear Dr Kapheim,

It is a pleasure to accept your manuscript entitled "Brain microRNAs among social and solitary bees" in its current form for publication in Royal Society Open Science.

Best regards,

on behalf of Dr Rees Kassen (Associate Editor) and Steve Brown (Subject Editor)
openscience@royalsociety.org

Appendix A

Associate Editor Comments to Author (Dr Rees Kassen):

This paper presents an interesting set of - largely negative - results around the putative association between miRNA regulation and sociality in bees. In brief, there doesn't seem to be any compelling link that miRNA evolution is linked to sociality. This is an interesting result and I commend the authors for revising their manuscript to reflect this conclusion more clearly. In light of these revisions, which are based on comments from an AE and three reviewers, I think the results are sound and the conclusions warranted. The one caveat, which deserves mention in the discussion, is the very limited sample size. This may be an inevitable consequence of the challenge of getting this kind of data but it is important to acknowledge explicitly, as the results could be different with broader sampling. Otherwise I am happy to recommend publication.

>>> We have now acknowledged the limited sample size in several places throughout the Discussion:

L288 - "We tested the hypothesis that miRNAs play a role in the evolution of eusociality via their regulatory effects on gene networks by comparing miRNA expression in a single bee from three eusocial and three solitary bee species that span three families. Our results provide very limited support for this hypothesis." (L288)

L300 – "Further investigation with deeper sampling and additional social and solitary species is necessary to determine if this miRNA is expressed exclusively in the brains of social species and how it may influence social behavior."

L327 – "A purging process could explain why there are large differences in the numbers of miRNAs detected in even closely related species (e.g., the two *Bombus* species), though this could also be an effect of limited sampling."